# A hemimetabolous wing development suggests the wing origin from lateral tergum of a wingless ancestor

Takahiro Ohde[1,2,3 ✉], Taro Mito [4] & Teruyuki Niimi [2,3]

The origin and evolution of the novel insect wing remain enigmatic after a century-long discussion. The mechanism of wing development in hemimetabolous insects, in which the first functional wings evolved, is key to understand where and how insect wings evolutionarily originate. This study explored the developmental origin and the postembryonic dramatic growth of wings in the cricket *Gryllus bimaculatus*. We find that the lateral tergal margin, which is homologous between apterygote and pterygote insects, comprises a growth organizer to expand the body wall to form adult wing blades in *Gryllus*. We also find that Wnt, Fat-Dachsous, and Hippo pathways are involved in the disproportional growth of *Gryllus* wings. These data provide insights into where and how insect wings originate. Wings evolved from the pre-existing lateral terga of a wingless insect ancestor, and the reactivation or redeployment of Wnt/Fat-Dachsous/Hippo-mediated feed-forward circuit might have expanded the lateral terga.

[1] Department of Applied Biosciences, Graduate School of Agriculture, Kyoto University, Kyoto, Japan. [2] Division of Evolutionary Developmental Biology, National Institute for Basic Biology, Okazaki, Japan. [3] Department of Basic Biology, School of Life Science, The Graduate University for Advanced Studies, SOKENDAI, Okazaki, Japan. [4] Division of Bioscience and Bioindustry, Graduate School of Technology, Industrial and Social Sciences, Tokushima University, Tokushima, Japan. ✉email: ohde.takahiro.4n@kyoto-u.ac.jp

nsects were the first animal group to achieve powered flight. The significance of this development for successful radiation of this animal group is unarguable, yet the evolutionary history of the wing remains unclear after a century-long debate. Both the location and the morphogenetic process of the origin of the insect wing remain elusive[1]. Regarding the former, the degree of contribution of three elements is under discussion: the pleuron and two subdomains of the tergum, i.e. the lateral and bona fide terga, which are derived from the proximal leg segment and the dorsal body wall in an ancestral crustacean, respectively[1–4]. Regarding the latter, the question pertains to whether the wing is a modification of a pre-existing structure or a de novo elaboration[1–3,5–7].

To disentangle these contrasting views regarding where and how the wing originates evolutionarily, it is essential to determine where and how wings originate developmentally in a pterygote insect, whose body plan is comparable to an ancestral wingless body plan. Although these issues have been well studied in the fruit fly *Drosophila melanogaster*, its derived holometabolous development is unsuitable for direct comparison with the development of apterygote ancestors, because the evolution of holometabolous development split the life cycle into two modules, and the continuity of the body plan from the larval to the adult stage has been largely lost[8]. In contrast, hemimetabolous insects show no drastic changes in the overall body plan between immature nymphs and adults, and a limited number of body parts, such as the wing and genital organs, change substantially in size and pattern during the transition to adulthood. Because the first winged insects were obviously hemimetabolous species, a dissection of hemimetabolous wing development at the molecular level would help map the developmental origin of wings in the insect body plan and deduce the changes in development that are responsible for the emergence of wings from a wingless body plan.

To achieve this goal, we investigate wing development in the two-spotted cricket *Gryllus bimaculatus*. In this study, we find that *apterous* (*ap*) induces *vestigial* (*vg*) at the dorsoventral boundary to form nymphal tergal margins in *Gryllus*. The *vg*-dependent lateral tergal margins are required for organizing the dramatic growth of wings during postembryonic development. We further show that three signaling pathways, the Wnt, Fat, and Hippo pathways, play a critical role in wind growth. Our results strongly support the conclusion that the *vg*-dependent lateral tergum is the developmental origin of the *Gryllus* wings, and that its subsequent growth is driven by the evolutionarily conserved Wnt/Fat-Dachsous/Hippo-mediated feed-forward (FF) circuit. This illustrates an evolutionary scenario in which either reactivation or redeployment of the FF circuit expanded the lateral terga of the wingless ancestor to achieve the first powered flight on Earth.

## Results

**Marker genes express in both the tergal and pleural regions.** To determine the developmental origin of wings, we first examined the mRNA expression patterns of *vestigial* (*vg*), *apterous* (*ap*), and *wingless* (*wg*) in *Gryllus* embryos. Given their central roles in wing development in *Drosophila*, these genes were selected as wing marker genes for finding structures homologous to wings in other insects and crustaceans[2,5,9–13]. *vg* in *Drosophila* embryos is the earliest marker of wing and haltere imaginal disc cells, although its role in determination of imaginal disc fate is unclear[14]. *vg* is required for the selective proliferation and identity formation of the prospective wing region of larval imaginal discs[14,15]. In late second instar larvae, Ap in the dorsal compartment of the wing disc induces both *vg* boundary enhancer (*vg*BE) activity and *wg*

expression at the dorsoventral (DV) boundary via the Notch pathway[16–18]. Subsequently, Wg from DV boundary cells, together with Decapentaplegic (Dpp) from anteroposterior (AP) boundary cells, activates *vg* quadrant enhancer (*vg*QE) to expand the wing compartment in the wing disc[15,19].

We initially examined the expression patterns of *wg*, *ap*, and *vg* in stages 6, 7, and 9 embryos of *Gryllus* (Fig. 1a). Tergal and pleural regions are subdivided during these stages[20]. In the thoracic tergal region, spots of *wg* mRNA expression appear on the posterior side in early stage 7 embryos and become more prominent in late stage 7 (Fig. 1b, c). Additional *wg* expression in lateral to posterior tergal edges is detected in stage 9 (Fig. 1d). *wg* is expressed in stripe patterns at the AP boundary of legs in the pleural region, as previously reported (Fig. 1b–d)[21]. Two *ap* orthologues in the *Gryllus* genome were classified to ApA and ApB from six amino acid residues unique to each group in the

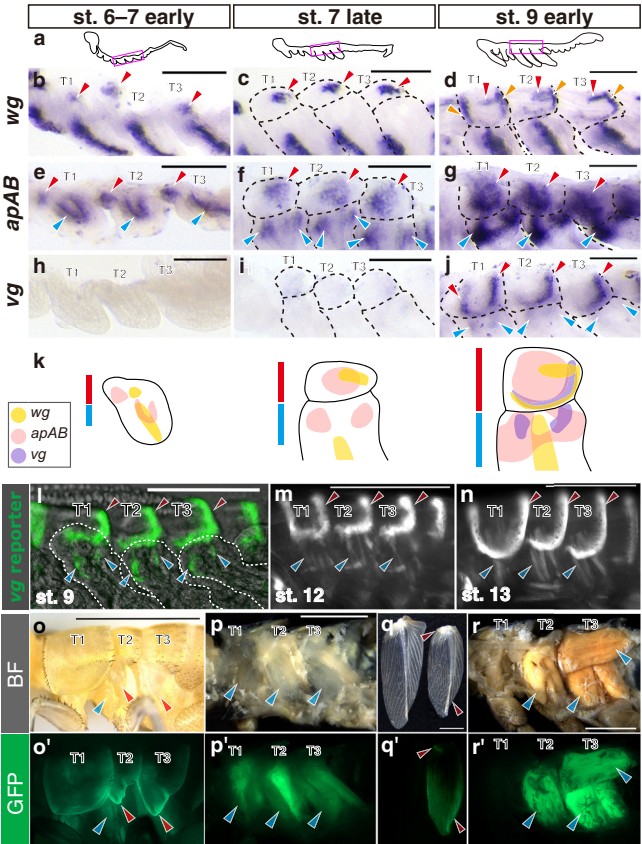

**Fig. 1 Expression pattern of *wg*, *apAB* and *vg* in thoracic segments of *Gryllus*. a** Schematics illustrate lateral views of *Gryllus* embryos in each stage. Magenta rectangle indicates the region shown in **b–j**. Dorsolateral regions of thoracic segments stained for *wg* (**b–d**), *apAB* (**e–g**) and *vg* (**h–j**) with in situ hybridization in stage 6 to early stage 7 (**b, e, h**), late stage 7 (**c, f, i**) and early stage 9 (**d, g, j**). Each image shows a representative result of at least three independent experiments with similar results. **k** Summary of gene expression pattern in meso- and metathoracic segments. Red and blue lines indicate tergal and pleural regions, respectively. **l–n** *vg* reporter gene signals in dorsolateral regions of thoracic segments in late embryos. **o–r** and **o′–r′** *vg* reporter gene signals in postembryonic stages. Dorsolateral region (**o, o′**) and thoracic musculature in median section (**p, p′**) of sixth instar nymph, and forewing (**q, q′**) and thoracic musculature in median section (**r, r′**) of adult. Crickets before cuticle coloration are shown for both nymph and adult. Red and blue arrowheads indicate expression in tergal and pleural regions, respectively. Scale bars are 100 μm in **b–j**, 250 μm in **l–n** and 3 mm in **o–r**, respectively.

homeodomain (Supplementary Fig. 1a)[22]. However, partial sequences we obtained were insufficient to separately analyze in situ expression of each gene (Supplementary Fig. 1b). We detected *apAB* signals in thoracic terga of stage 6 embryos that were maintained to stage 9 (Fig. 1e–g). *apAB* expression covers a broad area of thoracic terga, yet marginal cells lack expression. *apAB* expression is also detected in the thoracic pleural regions in addition to tergal cells. *apAB* is expressed in a U-shape pattern in stage 6, becomes separated into two areas in late stage 7, and forms connected areas, reshaping the U in stage 9 (Fig. 1e–g). Clear *vg* signals were first detectable at stage 9, in contrast to *wg* and *apAB* (Fig. 1h–j). Expression localized to tergal margins and two pleural spots (Fig. 1j). Marginal expression patterns of *wg* and *vg* in stage 9 embryos are both localized to the lateral to posterior regions of all thoracic segments, and also in the anterior region of prothorax (T1) (Fig. 1d, g, j, k).

*vg* is an embryonic marker of wing disc cells in *Drosophila*[14]. The wing disc includes a primordium of the wing, notum and pleural tissues. The formation of this compact imaginal disc is characteristic only in higher Diptera[23], and the fate of *vg*-expressing embryonic cells in insects other than *Drosophila* is unknown. In *Gryllus*, visualization of gene expression with in situ hybridization is limited to early embryos due to outer cuticle formation that causes strong non-specific staining in late embryos[21]. To track the fate of *vg*-expressing cells in *Gryllus*, we generated a *vg* reporter line by knocking-in the EGFP cassette at the upstream side of the *vg* coding sequence (Supplementary Fig. 2a–c). This reporter line (vg5′GFP) mimics *vg* mRNA expression in both tergal and pleural regions at stage 9 (Fig. 1l). During late embryonic development, the EGFP signal pattern in the pleural region is transformed from two spots to several cylindrical shapes (stage 12–), but the tergal region is maintained without significant change from earlier stages (Fig. 1l–n, Supplementary Fig. 1d–r). *vg* reporter signal in nymphs appears at the margins of wing pads while strong signaling disappears in the posterior tergal margin regions (Fig. 1o, o′). A signal was also detected in thoracic dorsoventral muscles (Fig. 1p, p′). *vg* reporter signals were detected in adult wings in a broad area at the distal region and at a spot in the proximal region (Fig. 1q, q′). We detected robust reporter activity in adult thoracic musculature including dorsal longitudinal and dorsoventral and pleural muscles that are indirect- and direct flight muscles, respectively (Fig. 1r, r′, Supplementary Fig. 3)[24]. These reporter expression patterns indicate that the tergal *vg*-expressing cells give rise to wings, while pleural *vg*-expressing cells comprise myoblasts and contribute to adult flight muscle formation.

**apAB induces *vg* to form a nymphal thoracic tergal margin**. To assess the role of *vg* and *apAB* in body plan formation of *Gryllus*, we generated CRISPR/Cas9-mediated mosaic knockouts for each gene. A loss of *wg* function likely causes no effects due to functional redundancy[25], and we focused on *vg* and *apAB* functions. We designed a sgRNA targeting *EGFP* as a negative control (Fig. 2a–e; Supplementary Table 1). Consistent with the expression pattern during embryogenesis, *vg* sgRNA/Cas9 injected individuals (*vg* crispants) exhibited a lack of tergal margins in both wingless (T1) and wing-bearing (T2 and T3) segments of first instar nymphs (Fig. 2f–h). Further, individuals without thoracic tergal margins displayed severely reduced wings in adults (Fig. 2i, j). Characteristic patterns in thoracic terga in *apAB* crispants, such as color, shape, and hairs were lost (Fig. 2k–m). Affected tergal areas lost their normal smooth surface, replaced by the rough surface seen in intersegmental membranes of wildtype insects (Fig. 2n). Loss of black color was also induced in lateral-posterior regions of the abdominal terga in *apAB* crispant nymphs (Fig. 2k). These nymphs did not survive to adulthood. In addition to tergal surface identity, the *vg*-dependent tergal margin structure appeared to be lost in

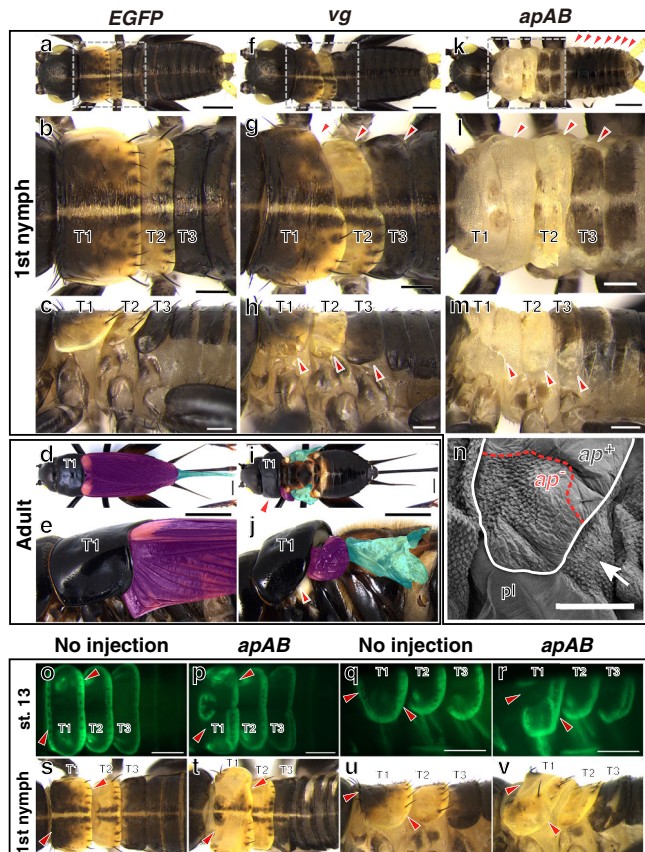

**Fig. 2 *apAB* induces *vg* at the DV boundary to form tergal margins.** Representative images of first instar nymphs (**a**–**c**, **f**–**h**, **k**–**m**) and female adults (**d**, **e**, **i**, **j**) injected with Cas9 protein and sgRNAs targeting *EGFP* (**a**–**e**), *vg* (**f**–**j**) and *apAB* (**k**–**m**) in early embryos from dorsal (**a**, **b**, **d**, **f**, **g**, **i**, **k**, **l**) and lateral (**c**, **e**, **h**, **j**, **m**) views. Boxed area in **a**, **f**, **k** is magnified in **b**, **g**, **l**, respectively. Arrowheads indicate regions affected by gene knockouts. Fore- and hindwings are shaded in magenta and cyan, respectively. Arrowheads indicate affected tergal regions. **n** A scanning electron micrograph of the dorsolateral region in mesothorax of an *apAB* mosaic knockout cricket. The tergum is outlined with a white line. Tergal surface with specific (*ap*−) and wildtype (*ap*+) phenotypes separated by red dotted line. The rough surface structure of *ap*− area resembles a soft intersegmental region (arrow). The same imaging was repeated for 10 independent *apAB* knockout crickets and similar phenotypes were observed. pl pleuron. *vg* reporter expression at st. 13 embryos (**o**–**r**) and phenotypes of first instar nymphs (**s**–**v**) of control and *apAB* crispant from dorsal (**o**, **p**, **s**, **t**) and lateral (**q**, **r**, **u**, **v**) views. Scale bars are 250 μm in **a**–**c**, **f**–**h**, **k**–**m**, **o**–**v**, 2 mm, **d**, **e**, **i**, **j** and 100 μm in **n**.

*apAB* crispant nymphs (Fig. 2k–n), indicating that *ap* regulates *vg* expression at the DV boundary, as observed in the *Drosophila* wing disc[16]. To assess the role of *apAB* in DV boundary formation further, we injected the *apAB* sgRNA/Cas9 into the vg5′GFP strain. The expression of the *vg* reporter at the DV boundary was partially lost after *apAB* sgRNA/Cas9 injection (Fig. 2o–r; Supplementary Fig. 4). A hatched nymph indicated the loss of the carinated margin structure and marginal bristles in regions where *vg* reporter expression was disturbed (Fig. 2s–v). These results demonstrated that *apAB* induces *vg* expression to form the nymphal tergal margin at the DV boundary.

**The LA region of tergum is essential for wing formation**. The severe defect in wing formation that follows the loss of nymphal

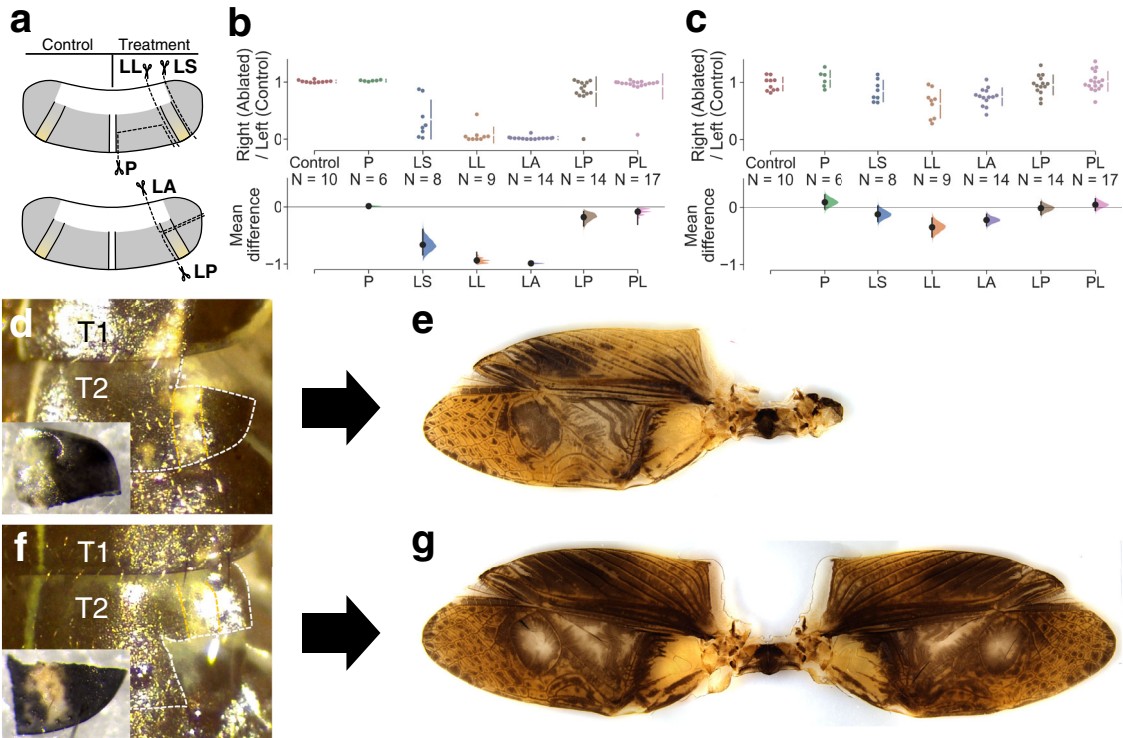

**Fig. 3 Identification of wing growth organizer in the lateral-anterior region of thoracic terga. a** Schematic illustration indicates ablated regions in the third instar nymph: P posterior, LS lateral-small, LL lateral-large, LA lateral-anterior, LP lateral-posterior. Effect of nymphal tergal ablations on wing (**b**) and articulation (**c**) sizes in adults. PL pleuron. Cumming plots indicate relative area of the treated right side to untreated left side (top) and mean differences of the values from control for wing (**b**) and articulation (**c**) regions (bottom). Source data are provided as a Source Data file. **d–g** Representative results of the ablation treatment. The right side of targa of the third instar nymph after the ablation of the LA (**d**) and LP (**e**) regions. The ablated tissues are shown in insets. Resultant adult wings of the same individuals are shown in **f** and **g**, respectively. Mesoterga and the yellowish line is outlined with white and yellow dotted lines, respectively.

tergal margins in *vg* crispants suggests that the tergal margin region associated with *vg* expression during embryogenesis is required for wing growth. During postembryonic development, lateral regions in T2 and T3 show dramatic exponential growth, while the rest the same segment exhibits linear growth (Supplementary Fig. 5). *vg* expression and function in *Gryllus* embryos are displayed in lateral and posterior margin regions of T2 and T3 terga, and this growth pattern suggests that the lateral region comprises the wing growth organizer. We ablated a part of the mesothoracic tergum in the third instar nymph to test this hypothesis and analyzed effects on adult wing size (Fig. 3a). Adult wing size is unaffected after posterior tergum ablation, while removal of lateral regions severely reduced wing size (P, LS, and LL in Fig. 3b). This effect is more prominent when larger regions of the lateral tergum are ablated (LS and LL in Fig. 3b). A similar size reduction was found in wing articulation, although the effect is small (Fig. 3c). We independently ablated anterior and posterior regions to further identify the lateral region critical to wing formation (Fig. 3a). We found that removal of the anterior region results in almost complete loss of the wing blade, whereas removal of the posterior region has almost no effect on wing size (LA and LP in Fig. 3b, d–g). Thus, the growth organizer of wing blades is located in the lateral-anterior (LA) region of terga. The effect of ablation of the LA region of a tergum is less prominent on the size of the wing articulation (Fig. 3c). The formation of an intact anteroposterior pattern after ablation of the lateral-posterior region indicates that the axis on the adult wing is secondarily formed during postembryonic development (Fig. 3g).

To confirm the major role of the tergal tissue in wing formation, we also ablated pleural tissues (Supplementary Fig. 6).

Ablation of the pleural tissues causes defects in the formation of epipleurites, such as the basalare and subalare, which are derived from the upper ends of the episternum and epimeron, respectively (Supplementary Fig. 6)[26]. However, loss of the pleuron had a negligible effect on the size of both the wing and the articulation areas (PL in Fig. 3b, c).

**Wnt, Fat, and Hippo signaling are required for wing growth.** To understand how *Gryllus* wings grow from the nymphal lateral terga, we next compared transcriptomes between lateral and central parts of terga (Fig. 4a, b). The steroid hormone 20-hydroxyecdysone (20E) promotes wing growth by increasing both cell number and cell size in lepidopteran insects[27,28]. We thus examined the expression level of two ecdysone responsive genes, *E74* and *E75*, as proxies for hemolymph ecdysone titer to determine the timing of tissue sampling for RNA-seq analysis[29,30]. *E74* and *E75* in thoracic segments showed expression peaks at 3- and 2.5-days post ecdysis to third instar (DPE), respectively (Supplementary Fig. 7). We selected 3 DPE as a growing stage since total RNA yield is highest among third instar nymphs, suggesting active transcription and protein synthesis (Supplementary Fig. 7).

We obtained raw reads from lateral and central regions of both T2 and T3 in 0 DPE and 3 DPE with a sufficient quality (Q30 > 95.87%; Supplementary Fig. 8a). We assembled a transcriptome from these reads with embryonic reads (BUSCO complete orthologues 95.6%; Supplementary Fig. 8b, c), and quantified expression level of each transcript. Transcriptomes display greater similarity within time point than within regions (Supplementary Fig. 9a). We analyzed differentially expressed genes (DEGs) between regions (Lateral versus Central) and

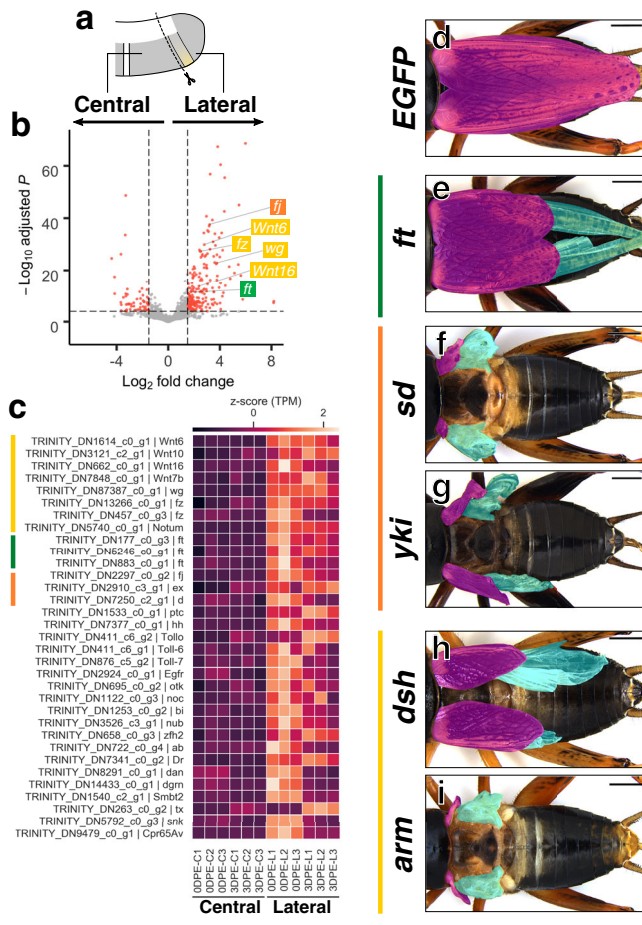

**Fig. 4 Wnt, Ft-Ds and Hippo signaling are required for the postembryonic wing growth in *Gryllus*. a** Schematics illustrates tergal regions used for the transcriptome comparison. Note that only T2 and T3 segments were used for this analysis. **b** A volcano plot shows differential expression of transcripts between central and lateral regions in T2 and T3 terga. Data after log-fold change shrinkage with the apeglm method is visualized. Significantly differentially expressed transcripts (adjusted $P < 10e-5$, fold change >1.5) are in red. A Wald test was used for significance testing and $P$ values were adjusted using the Benjamini and Hocheberg method according to DESeq2 software. Some representative transcripts that are highly expressed in the lateral region are labeled. **c** Heatmap indicating expression level of 33 candidate genes in each sample. Color code indicates z-score normalized transcripts per million (TPM). **d–i** Adult phenotypes after nymphal RNAi treatment. Fore- and hindwings are shaded in magenta and cyan, respectively. Scale bars are 2 mm.

region to identify upstream wing growth regulators (Fig. 4c). Notably, we found that genes involved in Wnt (Wnt ligands, the receptor, *frizzled*, and the Wnt ligand inactivator, *Notum*), Fat-Ds (the protocadherin, *fat* (*ft*)), and Hippo (the Golgi-kinase, *four-jointed*, the FERM domain gene, *expanded*, and the atypical myosin, *dachs* (*d*)) pathways show high expression in the lateral region. These pathways generate an FF circuit that plays a central role in the expansion of the wing compartment in *Drosophila* (Fig. 4b, c)[31]. In contrast, components of BMP signaling, another well-characterized signaling pathway that plays a pivotal role in the *Drosophila* wing disc growth and patterning[32], was not identified as a DEG between regions (Supplementary Fig. 10).

We selected 33 genes, including components of Wnt, Ft-Ds, and Hippo pathways, as candidate growth regulators and performed nymphal RNAi (nRNAi)-mediated functional screening to examine functions in *Gryllus* wing formation (Fig. 4c, Supplementary Table 2). We found specific phenotypes for six transcripts: two *ft* transcripts, *d*, the T-box transcription factor *optomotor-blind* (*omb*), *Epidermal growth factor receptor* (*Egfr*), *zinc finger homeodomain 2* (*zfh2*), and *abrupt* (*ab*). A BLAST-based comparison to the *Drosophila* official gene showed two *Gryllus ft* transcripts that showed effects on the wing size after RNAi treatment derive from a single *ft* locus (Supplementary Fig. 11). Among six genes with specific phenotypes, depletion of *ft* resulted in reduced wing size compared to control *EGFP* dsRNA injected crickets, suggesting a role of Ft-Ds pathway in the *Gryllus* wing growth (Fig. 4d, e). *omb* RNAi crickets exhibit irregular wing vein patterns around the AP boundary (Supplementary Fig. 12a–d). RNAi treatment of *zfh2* caused high lethality, but surviving adults commonly showed disorganized vein patterns (Supplementary Table 2; Supplementary Fig. 12e, f). *Egfr* RNAi crickets display small adult body sizes, as previously reported[33], but did not induce noticeable disproportionate effects on wing size (Supplementary Fig. 12g). We found that *ab* RNAi treatment causes similar small body phenotype (Supplementary Fig. 12g). Depletion of *d*, a regulator of the Hippo pathway, caused irregular male vein patterns and slightly smaller wings although this effect was not clear because of the modest impact (Supplementary Fig. 12h).

We targeted genes encoding effectors of Hippo pathway, *scalloped* (*sd*) and *yorkie* (*yki*), to further confirm the role of the Hippo pathway in wing growth. Depletion of *sd* caused a severe reduction in wing size (Fig. 4f; Supplementary Table 3). We failed to analyze the effect of *yki* dsRNA injections of third instar nymphs on wing growth due to lethality, but injection of sixth instar nymphs severely reduced wing size (Fig. 4g; Supplementary Table 3).

In contrast, we found no noticeable effect on wing formation after a single knockdown of each Wnt pathway component (Supplementary Table 2). This result may reflect functional redundancy of ligands and receptors reported in *Gryllus* and other species[25]. We targeted intercellular components of the canonical Wnt signaling pathway *disheveled* (*dsh*) and *armadillo* (*arm*) to clarify the involvement of the Wnt pathway (Supplementary Table 3). RNAi treatment for both genes resulted in severe reductions in wing size, showing the central role of the Wnt pathway in the *Gryllus* wing growth (Fig. 4h, i). Separate RNAi treatment against *ft*, *yki*, *sd*, *dsh* and *arm* resulted in the reduced size of both female ovipositor and antennae (Supplementary Fig. 13a, b). *d* RNAi crickets displayed curved and short ovipositors (Supplementary Fig. 13c). Ovipositor and antenna size indicate disproportional growth during postembryonic development, suggesting a shared role of Wnt/Ft-Ds/Hippo pathways in the growth regulation among disproportionally growing organs.

## Discussion

Regarding where the wing originates, our results suggest that wings evolved through a modification of the pre-existing lateral

between time points (0 DPE versus 3 DPE). We detected more DEGs between time points than between regions, consistent with the above findings (Supplementary Fig. 9b). As 3 DPE was presumed as a growing stage, transcripts highly expressed during this time were enriched with keywords such as "Developmental protein" and "Cell division", implying that genes involved in the growth of the lateral region were successfully identified (Supplementary Fig. 9c). DEGs in the lateral region were enriched with the keywords such as "DNA replication", "Cell cycle", and "Developmental protein", support our hypothesis that the lateral terga comprise the growth organizer in T2 and T3 (Supplementary Fig. 9c).

We then proceeded to develop a list of transcripts for RNAi-mediated functional screening. We focused on transcripts annotated as signaling molecules or transcription factors among DEGs more highly expressed in the lateral region than the central

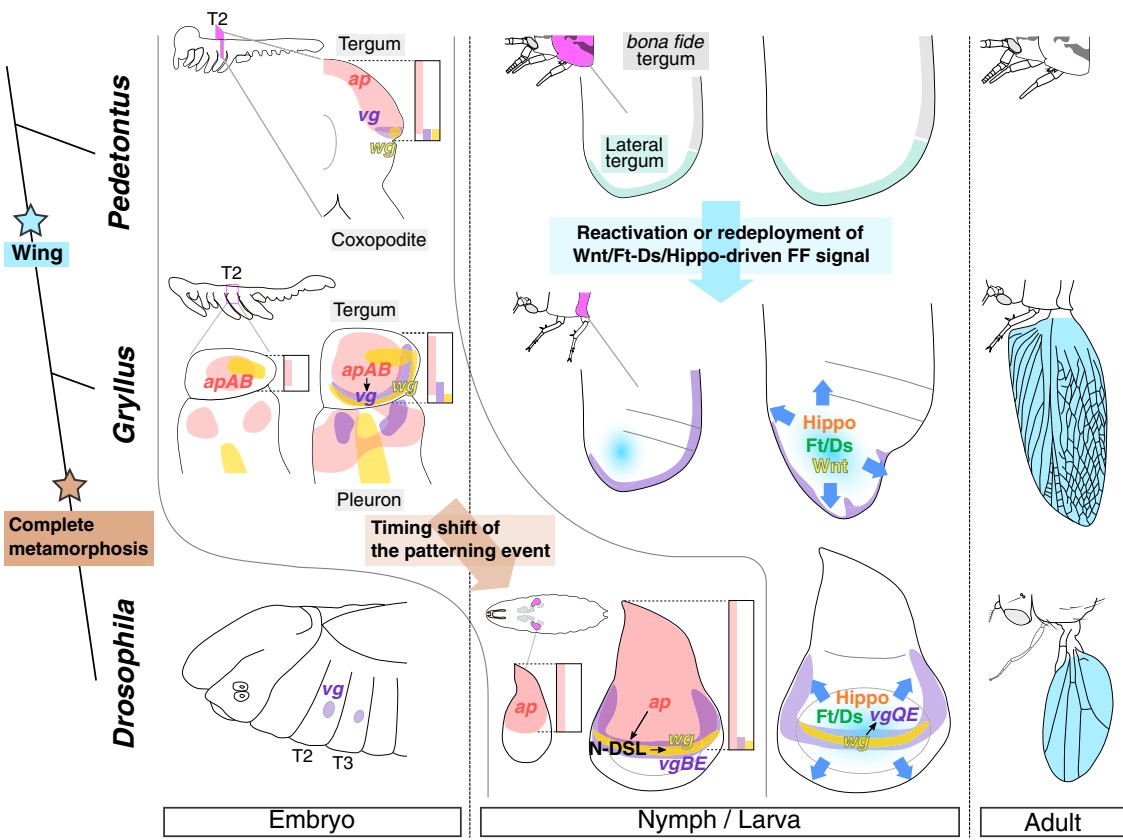

**Fig. 5 The development and evolution of the wing blade from the lateral tergal margin.** A resemblance of the dorsoventral distribution of *wg/ap/vg* between *Pedetontus*[10] and *Gryllus* indicates that nymphal terga are patterned in a similar fashion. The similar distribution is known in the *Drosophila* wing disc at the late second instar[18], which implicates a timing shift of the patterning event from embryonic to larval stage. *vg*BE activity in *Drosophila* is required to initiate expansion of the wing compartment via *vg* FF signaling transduced through actions of *vg*QE, Wg, and Ft-Ds and Hippo pathways[31]. Analogously, the *vg*-dependent tergal margin organizes the wing growth in *Gryllus*, and Wnt, Ft-Ds and Hippo pathways drive dramatic expansion during postembryonic development. A growth mechanism like that in *Drosophila* was employed at the lateral tergum of an ancestral apterygote insect to form the body wall extension required for the evolution of insect flight.

tergal margin tissue. The expression patterns of *apAB*, *wg*, and *vg* in *Gryllus* suggest the presence of structural continuity between apterygote and pterygote insects. These three genes exhibited a similar spatial distribution along the DV axis in the apterygote bristletail *Pedetontus unimaculatus* (Fig. 5)[10]. Thoracic terga are thus homologous between apterygote and pterygote insects. Modern hypotheses for wing origin suggest different scenarios of wing evolution, yet all hypotheses appear to agree on the major contribution of the tergum of the apterygote insect ancestor in the formation of wing blades[3,34–36]. Recent studies in crustaceans indicate that the tergal margin comprises two subdomains, namely the lateral tergum and posterior bona fide tergum, which have separate evolutionary origins. We showed that the lateral, but not posterior, region of the thoracic tergum organizes dramatic growth during the postembryonic development to form large wing blades in the hemimetabolous insect. The site of a wing growth organizer in *Gryllus* supports the tergal origin of wing blades and further specifies the origin at the lateral tergum, which derives from the proximal leg segment of an ancestral crustacean[3,4,37].

Regarding how the wing originates, the acquisition of a growth organizer at the pre-existing lateral tergum is a key developmental change for the transition from a wingless to a winged body plan in the scenario presented above. This study provides two critical insights on the evolution of the wing growth organizer. First, our data suggest that the mechanism to form a wing growth organizer is evolutionarily conserved among pterygote insects. In the wing

disc of the late second instar *Drosophila* larva, *ap* induces *vg* and *wg* at the DV boundary[18]. *vg* and *wg* organize growth of the wing field by sending an FF signal from the DV boundary[38]. In the *Gryllus* embryo, *apAB* is expressed across a broad region of terga, and *wg* and *vg* show localized expression in thoracic tergal margin cells. The *apAB* expression is required for *vg* expression at the DV boundary to form tergal margins. The LA part of the *vg*-dependent tergal margin organizes wing growth during postembryonic development. Such similarities in gene expression and the wing organizing function indicate that these developmental processes are equivalent between *Drosophila* and *Gryllus* although the timing is different (Fig. 5) and suggest that the mechanism to induce a wing growth organizer is deeply conserved among pterygote insects. Second, our study implies that some changes in a Wnt/Ft-Ds/Hippo pathway-mediated tissue growth mechanism had a significant impact on the modification of the lateral tergal tissue to form the wing. Our transcriptome and RNAi analyses pinpointed the critical roles of the Wnt/Ft-Ds/Hippo pathways in the postembryonic wing growth of *Gryllus*. Together with *vg*, these pathways make up the FF circuit to expand the wing field in the *Drosophila* imaginal disc[31]. Postembryonic depletion of *vg* represses wing growth in other hemimetabolous insects, *Oncopeltus fasciatus* and *Blattella germanica*[35,39]. These data suggest that the FF circuit is conserved between *Gryllus* and *Drosophila* as the core wing growth mechanism (Fig. 5). Intriguingly, recent studies have revealed that *vg*, *sd*, and *wg* exhibit localized expression and function in crustacean flat epidermal outgrowths,

such as the carapace, tergal edge, and coxal plate, although the homology of these flat outgrowths to insect wings remains controversial[1,2,6,7]. Because these three genes are components of the *vg*-dependent FF circuit, the FF circuit seems to have already been employed for the expansion of crustacean flat outgrowths before the emergence of insects and either the reactivation or redeployment of the "old" circuit in the lateral tergal cells extends the body wall of the wingless ancestor.

## Methods

**Animal husbandry**. *Gryllus bimaculatus* strain used in this study derives from the *gwhite* strain which is established and maintained in Tokushima University[40]. Cricket colonies were reared in an air-conditioned room at 28–30 °C with 12 L:12D photoperiod, and fed on artificial fish food (Spectrum Brands) and cat food (Purina One, Nestlé Purina Petcare). When precise staging is needed, we kept embryos and nymphs in an incubator (Panasonic) at 29 °C to minimize developmental stage variation at a time point. Embryo staging followed Donoughe and Extavour (2015)[41].

**In situ hybridization**. We searched *vg* and *ap* orthologs from *Gryllus* draft genome (GCA_017312745.1) with BLAST and identified partial nucleotide sequences (*vg*: LC589559, *apA*: LC589561 and *apB*: LC589562)[42]. The public sequence was used for *wg* (AB044713.1). Digoxigenin-labeled riboprobes were transcribed in vitro from 188 to 523 bp DNA fragments amplified with primers that have either a SP6 or T7 promoter sequence at the 5′ end (Supplementary Table 4). Embryos were dissected in PBS, and fixed in 4% formaldehyde overnight at 4 °C. Fixed embryos were washed in PTx (PBS, 0.1% TritonX-100), dehydrated in a MeOH series, and stored in 100% MeOH at −30 °C until use.

The following procedures were performed at room temperature otherwise described. Fixed embryos were rehydrated in a MeOH series, digested in 2 µg/ml Proteinase K for 5 min, and postfixed in 4% formaldehyde for 20 min. After prehybridization in hybridization buffer (50% deionized formamide, 5X SSC, 100 µg/ml heparin, 100 µg/ml yeast RNA, 0.1% TritonX-100, 0.1% CHAPS, 2% Roche blocking reagent) for more than an hour at 60–65 °C, 200–500 ng of a riboprobe were hybridized with gentle shaking for more than 60 h at 60–65 °C. Hybridized embryos were washed with a series of SSC buffer at the hybridized temperature, then with maleic acid buffer, and then blocked in 1.5% Roche blocking reagent (Merck & Co. #11096176001) for more than an hour. Blocked embryos were incubated in 1:2000 anti-Digoxigenin-AP (Merck & Co. #11093274910) overnight at 4 °C. After several washes with 1.5% Roche blocking reagent and maleic acid buffer, color was developed with NBT/BCIP solution.

**Size measurement**. We collected all samples from a single colony for minimizing variation from genetic and environmental background for assessment of area measurement in central and lateral regions in thoracic segments. We kept all hatched first instar nymphs in a plastic cage, and collect six or seven individuals after each molting. Each sex was separately collected from the fourth instar nymph to the adult. Nymphs and adults were digested in lactic acid overnight at 65 °C after cutting off unnecessary body parts, and remaining tergal cuticles of thoracic segments were mounted on glass slides. We mounted nymph specimens in Hoyer's media. Adult specimens were washed twice in 100% EtOH and mounted in EUKITT neo (O. Kindler ORSAtec). Images of mounted slides were captured with a DFC7000 T equipped with a M165 FC (Leica Microsystems) and analyzed with Fiji (ImageJ2, v2.0.0).

**CRISPR/Cas9-mediated genome editing**. We assembled oligonucleotides and transcribed and purified single-guide (sg) RNAs with a precision gRNA synthesis kit (ThermoFisher Scientific #A29377) according to the manufacturer's instruction. Oligonucleotides used for DNA template assembly are shown in Supplementary Table 4. Synthesized sgRNAs were aliquoted in small volumes and stored at −80 °C until use. For microinjection, eggs laid within 1–2 h were collected from wet paper towels, and pieces of cotton layered in plastic dishes and were aligned in wells in a 2% agarose gel after a brief wash in tap water. A pulled glass capillary connected to a Femtojet microinjector (Eppendolf) was used to inject small droplets of solution into eggs. Materials were injected within 4 h after egg oviposition.

For somatic gene knockouts, we injected 100 ng/µl sgRNA designed for each gene, and 500 ng/µl Alt-R S.p. Cas9 Nuclease V3 (Integrated DNA Technologies #15596018). Non-homologous end joining (NHEJ)-mediated gene knock-in is performed as previously described for generating the *vg* reporter line[43,44]. The donor plasmid was generated by integrating a partial DsRed sequence as sgRNA target and EGFP expression cassette driven by *Gryllus* cytoplasmic actin promoter[45]. A sgRNA targeting 5′ region of the *vg* protein-coding sequence was designed with a partial sequence obtained by BLAST search against the *Gryllus* genome assembly. We injected a solution containing 40 ng/µl of sgRNAs targeting the *vg* upstream site and the donor plasmid, 100 ng/µl of the donor plasmid, and 100 ng/µl of Cas9 mRNA transcribed from the linearized MLM3613 plasmid

(Addgene #42251)[46]. GFP-positive individuals were crossed to wildtype, and individuals with GFP signal were selected for establishing the *vg* reporter line.

**Tergum ablation**. The ablation was performed on third instar nymphs within 24 h of molting. Nymphs were anesthetized on ice, and a targeted region of mesotergum on the right-hand side was ablated with a pair of spring scissors. The yellowish line is used as a landmark for ablating different sizes of tissue (i.e. LS and LL in Fig. 3a). Ablated nymphs were separately kept in a plastic cup until eclosion, and mesoterga of adults were digested with lactic acid and mounted on glass slides for area measurement. Cumming plots were created with DABEST (v0.3.1)[47].

**Quantitative PCR (qPCR)**. Nymphs newly molted within 2 h to the third instar were periodically collected from a colony and separately kept in small plastic cups at 29 °C. Three to four nymphs were moved on ice for anesthetization at each time point, and thoracic segments without an alimentary canal were dissected in ice-cold PBS, collected individually in TRIzol (Thermo Fisher Scientific #15596018), and stored at −80 °C until use. Total RNA was extracted according to manufacturer's instructions, RNA pellets were resuspended in 30 µl of distilled water and quantified with a Nanodrop 2000 spectrophotometer (Thermo Fisher Scientific). We used 500 ng of total RNA for cDNA synthesis using ReverTra Ace qPCR RT Master Mix with gDNA Remover (Toyobo # FSQ-301), and 1 µl of ten-fold diluted cDNA in 10 µl qPCR reaction with THUNDERBIRD SYBR qPCR Mix (Toyobo #QPS-101). A Thermal Cycler Dice Real-Time System II (Takara Bio) was used for qPCR reaction, and we calculated relative gene expression level with the delta Ct method.

**mRNA-sequencing analysis**. Nymphs molted within 3 h to the third instar were periodically collected from a colony and separately kept in small plastic cups at a 29 °C incubator. Zero and 3 DPE samples were anesthetized on ice within 10 min to 2.5 h and 69 to 73 h post ecdysis, respectively. Lateral and central tergal parts of both meso- and metathorax were dissected on a paper towel on ice, and then washed in PBS. Dissected tissues from 25 nymphs for a sample were collected in TRIzol and stored at −80 °C until use. Total RNA was extracted following the manufacturer's instructions and was further purified with a RNeasy MinElute spin column (QIAGEN #74204). We also extracted total RNA from embryos incubated at 29 °C at days 2.5, 3, 3.5, 4.5, 5.5 and 6 after egg laying. Total RNA from terga and embryos were eluted from a column with 20 µl of RNase-free water and submitted to GENEWIZ and Filgen standard RNA-seq service (GENEWIZ) that provided a standard Illumina mRNA library and generated 150 bp pair-end reads on an Illumina HiSeqX and HiSeq4000 (Illumina), respectively.

We used trimmed reads longer than 25 bp after adaptor sequence removal by Cutadapt (v2.9) for the following sequence analysis[48]. We assembled all reads from both tergum and embryo samples together with Trinity (v2.8.4), and assessed quality of assembly with BUSCO (v4.0.5)[49,50]. After quantification of the expression level of each transcript with Salmon (v1.0.1)[51], DEGs were statistically identified with DESeq2 (v1.26.0)[52]. Functional annotation of assembled transcripts was performed with both Trinotate (v3.1.0) and BLASTX (v2.9.0) against the latest *Drosophila* protein sequences (r6.32). The FlyBase ID of the best BLAST hits in the search against *Drosophila* database was assigned to each transcript and used for gene set analysis with DAVID (v6.8)[53]. Data were visualized with EnhancedVolcano (v1.7.10), Python3 (v3.7.7) and R (v3.6.3)[54,55].

**Nymphal RNAi**. For the functional screening, we used a consensus sequence created from all isoforms for a target gene in the transcriptome to design a double-stranded RNA (dsRNA). The specificity of dsRNA was assessed by BLAST search to both the transcriptome assembled in this study and the genome[42]. RNA was transcribed in vitro from a PCR product as a template. PCR primers are listed in Supplementary Table 4. Transcribed RNA was digested with DNase I, then purified with a standard phenol/chloroform extraction protocol. After annealing, dsRNA was aliquoted and stored at −80 °C until use. One microliter of dsRNA was injected into the ventral side of the intersegmental membrane between second and third thoracic segments with a pulled glass capillary. Nymphs injected with the same dsRNA were kept in a plastic cage until reaching adulthood.

**Image analysis**. Epifluorescent/confocal/scanning electron microscopy images were taken with M165 FC and LAS X (v3.4.1) (Leica microsystems)/A1R MP and NIS-Elements AR (v4.13) (Nikon)/VHX-D500 (KEYENCE), respectively. Images were processed with either Fiji (ImageJ2, v2.0.0) or GIMP (v2.10) and assembled and annotated with Inkscape (v1.0 beta).

**Reporting summary**. Further information on research design is available in the Nature Research Reporting Summary linked to this article.

## Data availability

Nucleotide sequences of *Gryllus vg*, *apA*, and *apB* orthologues and the RNA-seq data characterized and generated in this study are deposited to DDBJ/EBI/NCBI database under accession numbers LC589559, LC589561, LC589562 and PRJDB10701, respectively. Source data are provided with this paper.

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

## Acknowledgements

We thank Drs. Tetsuya Bando and Sumihare Noji for their help in preliminary analysis of *Gryllus vg* and *sd* function. We thank Dr. Toshiya Ando, and Dr. Takaaki Daimon and his lab members for helpful discussion. We also thank Drs. Yuji Matsuoka, Takahito Watanabe, Yohei Katoh, Taro Nakamura, Shinichi Morita, Hajime Ono, Miki Sugimoto, and Mr. Takahisa Yamashita for technical supports. The computational resource for the RNA-seq analysis was provided by NIG supercomputer system. This study was supported by JSPS KAKENHI (16K18825 and 19H02970 for T.O., 16H02596 for T.N.).

## Author contributions

T.O.—conceptualization, methodology, formal analysis, investigation, writing—original draft preparation, visualization, project administration, funding acquisition; T.M.—methodology, validation, resources, writing—review and editing; T.N.—conceptualization, validation, resources, writing—review and editing, funding acquisition.

## Competing interests

The authors declare no competing interests.
