## [Peer Review File · Nature Communications]

A hemimetabolous wing development suggests the wing origin from lateral tergum of a wingless ancestorREVIEWER COMMENTS

Reviewer #1 (Remarks to the Author):

The paper combines four experimental approaches to characterize and functionally dissect wing development in *Gryllus*. 1) The authors provide a careful description of *wg*, *ap*, and *vg* expression. In particular, they use CRISPR/cas9 mediated genome editing to produce a *vg*-EGFP reporter line which allows the detection of tergal, pleural and wing *vg* expression after formation of cuticle. This description is very beautiful and valuable for future studies, but has no immediate mechanistic implications, at least the authors are not mentioning implications within the paper. 2) For the first time in a hemimetabolous insect the authors study wing development by generating genetic mosaics. Using CRISPR/cas9 they induce *vg* and *ap* clones. The phenotypes suggest that the *vg* mutant clones lead to the lack of tergal margins in T1-T3 and severe reduction of wing tissue. The effect of *ap* KD on wing development is not explicitly mentioned. The observed transformation is only asserted without further explanation. 3) The authors mechanically (using scissors) ablate tergal tissue. This shows that the wing organizer localizes to the anterior lateral regions of the tergum. 4) The authors search for differentially expressed genes in lateral versus central tergal regions. This approach reveals that Wnt, Ft-Ds and Hippos signaling promote wing growth. Together this paper presents a remarkable number of new approaches and observations on wing development in a hemimetabolous insects. It clearly localizes the wing organizer to the anterior lateral margin of the tergum. These results will further clarify the discussion on the origin of insect wings. In summary, this is an excellent study which should be of interest to many readers working in the field of evolutionary developmental biology.

Minor comments, suggestions and questions, which should be addressed:

- The authors should make a clear statement about the function of *ap* in *Gryllus* wing development. How is the dorsoventral compartment function of *ap* in *Drosophila* related to *ap* function in *Gryllus*?
- In general: Does the expression and functional analysis provide information on axial patterning of the wing (origin of AP and DV axis, potential compartment boundaries)?
- In the discussion section the authors should say more about the relevance of their data for the current debate about the origin of insects wings.

Reviewer #2 (Remarks to the Author):

The aim of the study is to elucidate the evolutionary origins of insect wings. This is an interesting question in evolutionary biology. However, the study does not compellingly deliver on this objective. I think the findings are publishable in a more focused journal but need to be re-framed. Here, the insight gained into the origins of insect wings is compromised by aspects of the experimental approach which I elaborate below. Due to the nature of these limitations, I do not feel the authors would be able to enact a revision that brings the paper to a suitable standard of broad and novel interest. I therefore recommend rejection.

The authors examine the timing and patterns of expression of a panel of candidate wing development genes in a hemimetabolous insect, the cricket *Gryllus bimaculatus*. They use a variety of techniques to visualise expression patterns, and then assess what happens to wing morphology when gene activity is disrupted through knock down/knock out. They find that pathways involved in wing development in *Drosophila melanogaster* (a holometabolous insect) are involved in varying ways in wing development of *G. bimaculatus*. Comparison against a more basal, wingless ancestor, the bristletail *Pedetontus unimaculatus*, implicates thoracic lateral tergal

regions in the evolution of wing development. The study is technically sound and the results displayed to a high standard. The ablation experiment illustrated in Figure 3 is particularly compelling.

On line 49, the authors argue that it is less suitable to compare holometabolous wing development with apterygote ancestors (e.g. the bristletail) than it is to compare hemimetabolous wing development to such ancestors to understand insect wing evolution. This logic is sound. However, the study then interrogates the spatiotemporal dynamics of *D. melanogaster* candidate wing development genes in the hemimetabolous subject, *G. bimaculatus*. This inadvertently skews subsequent inference – while knocking down/out *D. melanogaster* homologues can reveal their involvement in *G. bimaculatus* wing development pathways, the study is by design omitting genes or pathways that may be unique to hemimetabolous insects. It is unclear why the authors did not start with, or also include, other genes that have been implicated in other hemimetabolous insects, such as *Oncopeltus fasciatus* and *Blattella germanica* (e.g. Scr; Elias-Neto and Belles, Roy Soc Open Sci doi.org/10.1098/rsos.160347).

An attempt to rectify this problem is presented later in the paper by performing RNAseq using *G. bimaculatus* embryonic tissues, but this provides limited resolution of the issue and forms a more minor component of the study. It is valid to say that the study can inform us about the action of these particular core genes (*wnt*, *vg*, etc.) across representatives of three key groups (apterygote, hemimetabolous, holometabolous), but suggesting this informs the dual origin hypothesis or more generally the evolutionary origin of hemimetabolous insect wings (e.g. title) stretches the findings too far. While it must be appreciated that a formal comparative analysis, e.g. examining gene family expansions, or other standard comparative approaches, was not the object of the current study, that sort of analysis is needed to answer the questions proposed at the beginning of the Introduction. A more accurate statement of the study's goals is on line 60 – investigating the genetic mechanisms of how wings extend from nymphal tergae. While such data are a valuable resource, they do not in my view constitute a novel, broad advance that will influence the direction of the field.

REVIEWER COMMENTS

Reviewer #1 (Remarks to the Author):

The paper combines four experimental approaches to characterize and functionally dissect wing development in *Gryllus*. 1) The authors provide a careful description of *wg*, *ap*, and *vg* expression. In particular, they use CRISPR/cas9 mediated genome editing to produce a *vg*-EGFP reporter line which allows the detection of tergal, pleural and wing *vg* expression after formation of cuticle. This description is very beautiful and valuable for future studies, but has no immediate mechanistic implications, at least the authors are not mentioning implications within the paper. 2) For the first time in a hemimetabolous insect the authors study wing development by generating genetic mosaics. Using CRISPR/cas9 they induce *vg* and *ap* clones. The phenotypes suggest that the *vg* mutant clones lead to the lack of tergal margins in T1-T3 and severe reduction of wing tissue. The effect of *ap* KD on wing development is not explicitly mentioned. The observed transformation is only asserted without further explanation. 3) The authors mechanically (using scissors) ablate tergal tissue. This shows that the wing organizer localizes to the anterior lateral regions of the tergum. 4) The authors search for differentially expressed genes in lateral versus central tergal regions. This approach reveals that Wnt, Ft-Ds and Hippo signaling promote wing growth.

Together this paper presents a remarkable number of new approaches and observations on wing development in a hemimetabolous insects. It clearly localizes the wing organizer to the anterior lateral margin of the tergum. These results will further clarify the discussion on the origin of insect wings. In summary, this is an excellent study which should be of interest to many readers working in the field of evolutionary developmental biology.

Minor comments, suggestions and questions, which should be addressed:

[Remark 1-1]

The authors should make a clear statement about the function of *ap* in *Gryllus* wing development. How is the dorsoventral compartment function of *ap* in *Drosophila* related to *ap* function in *Gryllus*?

[Response 1-1]

We appreciate the constructive suggestion provided by the reviewer. We addressed this point by generating *apAB* crispants in the *vg* reporter strain and found that *apAB* was required for *vg* expression at the DV boundary to form nymphal tergal margins. We added this result to Fig. 2o–v and Supplementary Fig. 4.

[Remark 1-2]

In general: Does the expression and functional analysis provide information on axial patterning of the wing (origin of AP and DV axis, potential compartment boundaries)?

[Response 1-2]

We obtained limited information about the formation of the wing axis in this study. In our RNAseq analysis, we selected all known wing-patterning genes from the DEG list for the functional screening (Supplementary Table 2). Among these candidates, only omb RNAi caused defects in wing axis formation, as described in Line 361 and shown in Supplementary Figure 12.

Mechanical ablation of the lateral–posterior region of the tergum resulted in the formation of an intact axis (Fig. 3g). This indicates that the wing AP axis is formed secondarily. We accordingly added a pertinent sentence to Line 169.

[Remark 1-3]

In the discussion section the authors should say more about the relevance of their data for the current debate about the origin of insects wings.

[Response 1-3]

We revised the Introduction and Discussion based on the current discussion about the origin of the insect wing. We mainly used the framework shown in Tomoyasu (2021) to clarify our standpoint regarding where and how the insect wing originated.

Reviewer #2 (Remarks to the Author):

The aim of the study is to elucidate the evolutionary origins of insect wings. This is an interesting question in evolutionary biology. However, the study does not compellingly deliver on this objective. I think the findings are publishable in a more focused journal but need to be re-framed. Here, the insight gained into the origins of insect wings is compromised by aspects of the experimental approach which I elaborate below. Due to the nature of these limitations, I do not feel the authors would be able to enact a revision that brings the paper to a suitable standard of broad and novel interest. I therefore recommend rejection.

The authors examine the timing and patterns of expression of a panel of candidate wing development genes in a hemimetabolous insect, the cricket *Gryllus bimaculatus*. They use a variety of techniques to visualise expression patterns, and then assess what happens to wing morphology when gene activity is disrupted through knock down/knock out. They find that pathways involved in wing development in *Drosophila melanogaster* (a holometabolous insect) are involved in varying ways in wing development of *G. bimaculatus*. Comparison against a more basal, wingless ancestor, the bristletail *Pedetontus unimaculatus*, implicates thoracic lateral tergal regions in the evolution of wing development. The study is technically sound and the results displayed to a high standard. The ablation experiment illustrate in Figure 3 is particularly compelling.

[Remark 2-1]

On line 49, the authors argue that it is less suitable to compare holometabolous wing development with apterygote ancestors (e.g. the bristletail) than it is to compare hemimetabolous wing development to such ancestors to understand insect wing evolution. This logic is sound. However, the study then interrogates the spatiotemporal dynamics of *D. melanogaster* candidate wing development genes in the hemimetabolous subject, *G. bimaculatus*. This inadvertently skews subsequent inference – while knocking down/out *D. melanogaster* homologues can reveal their involvement in *G. bimaculatus* wing development pathways, the study is by design omitting genes or pathways that may be unique to hemimetabolous insects.

[Response 2-1]

We appreciate that the reviewer found the results reported in this study to be of high standard. We admit that the relationship between this study and current questions on the wing origin was ambiguous in the initial version of the manuscript. We revised both the Introduction and the Discussion to clarify the current questions and insights gained from this study.

The rationale behind studying the three wing marker genes was to identify candidate wing homologs in an ancestral insect body plan. Although this issue is important for understanding the location of the evolutionary origin of the insect wing, it remains unclear to date because of technical limitations regarding

manipulations in insects with a relatively ancestral body plan, such as hemimetabolous insects. We believe that the generation of the vg reporter line first revealed the fate of candidate wing homologs in early development and the major contribution of lateral terga as the wing origin, in combination with the subsequent functional analysis and the ablation experiment.

[Remark 2-2]

It is unclear why the authors did not start with, or also include, other genes that have been implicated in other hemimetabolous insects, such as *Oncopeltus fasciatus* and *Blatella germanica* (e.g. Scr; Elias-Neto and Belles, Roy Soc Open Sci doi.org/10.1098/rsos.160347).

[Response 2-2]

We appreciate the constructive suggestion provided by the reviewer. We agree that an analysis of Scr would help to determine the anatomical identity of the wing and its serial homologs. In fact, we attempted to knock down an *Scr* ortholog in *Gryllus* via RNAi; unfortunately, we failed to obtain clear knockdown effects for unknown reasons. We attempted to use the consecutive injection method reported in *Oncopeltus* and *Blatella*, as well as different sequences of dsRNA, without success. Hopefully, we will be able to overcome this issue and report the results in a future study.

[Remark 2-3]

An attempt to rectify this problem is presented later in the paper by performing RNAseq using *G. bimaculatus* embryonic tissues, but this provides limited resolution of the issue and forms a more minor component of the study. It is valid to say that the study can inform us about the action of these particular core genes (*wnt*, *vg*, etc.) across representatives of three key groups (apterygote, hemimetabolous, holometabolous), but suggesting this informs the dual origin hypothesis or more generally the evolutionary origin of hemimetabolous insect wings (e.g. title) stretches the findings too far.

[Response 2-3]

After characterizing the origin of the wings (i.e., lateral terga), we performed an RNAseq analysis to address how wings originate. In this section of the manuscript, we aimed to perform a comprehensive search for genes involved in postembryonic wing growth. We did not discover taxon-specific players in wing growth in this analysis; rather, we identified a deep conservation in the wing growth mechanism between *Gryllus* and *Drosophila*. We believe that this finding is important because it suggests that the core mechanism underlying the expansion of wings is conserved among phylogenetically distant pterygote insects, and we could extrapolate that the acquisition of this developmental process is key for the wingless-to-winged transition.

[Remark 2-4]

While it must be appreciated that a formal comparative analysis, e.g. examining gene family expansions, or other standard comparative approaches, was not the object of the current study, that sort of analysis is needed to answer the questions proposed at the beginning of the Introduction. A more accurate statement of the study's goals is on line 60 – investigating the genetic mechanisms of how wings extend from nymphal tergae.

[Response 2-4]

We revised the Introduction to clarify the current questions regarding the origin of the wing and our goal to understand where and how wings originate in a hemimetabolous insect. We agree that other approaches (such as genomics) will help to understand wing evolution in the future.

[Remark 2-5]

While such data are a valuable resource, they do not in my view constitute a novel, broad advance that will influence the direction of the field.

[Response 2-5]

We believe that this study will unravel the current views regarding where and how the wing originates and will affect the direction of future studies by providing evidence on (1) the developmental origin of hemimetabolous wings from the lateral terga, (2) the structural continuity between apterygote and pterygote insects, and (3) the possible key role of the Wnt/Ft-Ds/Hippo-mediated FF circuit in the wingless-to-winged transition. These findings constitute a solid basis for a formal comparative analysis, for instance from developmental and genomics approaches, between apterygote and pterygote insects.

REVIEWERS' COMMENTS

Reviewer #1 (Remarks to the Author):

The authors have responded to all my queries. They include now the KD of apterus and they have extensively rewritten the manuscript to more fully consider the current state of the discussion on the origin insect wings.

As mentioned earlier and in agreement with the other reviewer the data presented in this paper are very solid and use cutting-edge KD and transgenesis technologies. In this sense the paper also presents a significant technical advance. I partially agree with the other reviewer's view that the candidate gene approach used in this paper, is not appropriate to discover the molecular changes responsible for the evolution of wings. However, the current paper adds a data set important for all future work on this topic. It also introduces *Gryllus* as a functional system for dissecting molecular mechanisms wing evolution.

Reviewer #3 (Remarks to the Author):

In this study, the authors examine the developmental origins of insect wings, focusing on a hemimetabolous model *Gryllus bimaculatus*. The sheer amount and quality of data is outstanding, from standard RNAi insights to creation of transgenic vg-reporter line and CRISPR/Cas-mediated knockouts to transcriptome analysis. The generated results clearly establish that the wing organizer is localized the anterior lateral margin of the tergum, which is an exciting new finding of general significance. Furthermore, the authors also identified six upstream wing growth regulators belonging to Wnt, Fat-Ds, and Hippo pathways. In a broader sense, this work establishes *Gryllus* as a new model for studying wing evolution in hemimetabolous insects, which is an important step forward as most of the previous insights were generated using holometabolous species such as *Drosophila*, *Tribolium*, and *Tenebrio*. Hence, I recommend publication with the following suggestions to further improve text and data presentation:

1) In terms of the overall tone and conceptual positioning of this study, the authors undersell their own data.

The previous critical work on wing evolution in the past decade was based mainly on studies in *Tribolium* and partially from *Tenebrio*, both of which are undergoing derived holometabolous development (Hechtel-Clark et al. 2013; Linz&Tomoyasu 2018). Complementing previous work in *Oncopeltus* (Medved et al 2015), this study in *Gryllus* provides powerful new insight into earlier stages of wing evolution in more basal, hemimetabolous lineages. As a matter of fact, present results actually provide the evidence in support of step 1 (out of 4-step scenario; Fig. 6) proposed by Medved et al – mainly how a set of dorsally derived outgrowths evolved in early hemimetabolous lineages.

Recommended changes:

- in both abstract and discussion highlight the *Gryllus* results as evidence for the above mentioned step 1, which in turn provides a critical connection between "dorsal flaps" observed in fossils and present-day wings

- similarly, highlight the indispensable insight from hemimetabolous species such as *Oncopeltus* and *Gryllus* (and to lesser degree *Periplaneta* & *Blattella*) for understanding early wing evolution. In other words, whereas experiments in *Tenebrio* and *Tribolium* were important for establishing the combinatorial nature of present-day wings, to understand the details of such multi-step evolutionary process we need to shift focus to species that undergo hemimetabolous mode of development (as this mode evolved earlier).

2) In Fig 1, panel K – change the color to provide better contrast for a diagram of genes expression (perhaps: red, light blue, dark blue; matching the distinction between tergal and pleral regions).

3) In Fig 3, reverse the order of panels: start with a), then present d-e and f-g panels (to illustrate some of the phenotypes), then finish with b-c panels to summarize all the results.

Minor comment:

- in references, articles listed as 5-6 and 41-44 are missing (either omit or add to the list).